

# Change-point detection in wind turbine SCADA data for robust condition monitoring with normal behaviour models

Simon Letzgus

Technische Universität Berlin, Machine Learning Group; Straße des 17. Juni 135, 10623 Berlin, Germany

**Correspondence:** simon.letzgus@tu-berlin.de

**Abstract.** Analysis of data from wind turbine supervisory control and data acquisition (SCADA) systems has attracted considerable research interest in recent years. The data is predominantly used to gain insights into turbine condition without the need for additional sensing equipment. Most successful approaches apply semi-supervised anomaly detection methods, also called normal behaivour models, that use clean training data sets to establish healthy component baseline models. However,

one of the major challenges when working with wind turbine SCADA data in practice is the presence of systematic changes in signal behaviour induced by malfunctions or maintenance actions. Even though this problem is well described in literature it has not been systematically addressed so far. This contribution is the first to comprehensively analyse the presence of change-points in wind turbine SCADA signals and introduce an algorithm for their automated detection. 600 signals from 33 turbines are analysed over an operational period of more than two years. During this time one third of the signals are affected by

change-points. Kernel change-point detection methods have shown promising results in similar settings but their performance strongly depends on the choice of several hyperparameters. This contribution presents a comprehensive comparison between different kernels as well as kernel-bandwidth and regularisation-penalty selection heuristics. Moreover, an appropriate data pre-processing procedure is introduced. The results show that the combination of Laplace kernels with a newly introduced bandwidth and penalty selection heuristic robustly outperforms existing methods. In a signal validation setting more than 90%

of the signals were classified correctly regarding the presence or absence of change-points, resulting in a F1-score of 0.86. For a change-point-free sequence selection the most severe 60% of all CPs could be automatically removed with a precision of more than 0.96 and therefore without a significant loss of training data. These results indicate that the algorithm can be a meaningful step towards automated SCADA data pre-processing which is key for data driven methods to reach their full potential. The algorithm is open source and its implementation in Python publicly available.





# 1 Introduction

Wind energy plays a major role in decarbonisation of energy systems around the world. It has developed into a mature technology over the past decades and its levelised cost of electricity (LCOE) has reached a competetive level (IRENA (2019)). At the same time costs for operation and maintenance (O&M), which account for approximately one quarter of the LCOE, have

seen only minor reductions (IRENA (2019)). An effective strategy to further reduce O&M costs is to switch from a scheduled maintenance scheme to condition-based maintenance. Under such a scheme maintenance decisions are based on information about the turbine's actual condition rather than on periodic inspections. The necessary information can be acquired through dedicated condition monitoring (CM) systems which can be for instance vibration-, oil- or acoustic emission-based (for a comprehensive review of state-of-the-art wind CM systems please refer to (Coronado and Fischer (2015))). On the other hand, each

wind turbine is equipped with a variety of sensors in its supervisory control and data acquisition (SCADA) system. Utilisation of operational SCADA data for CM has attracted considerable research interest since it provides insights with no need for additional equipment. A wide range of methods have proven to be able to detect developing malfunctions at an early stage, often months before they resulted in costly component failures (see e.g. Zaher et al. (2009), Schlechtingen and Santos (2011), Bangalore et al. (2017), Bach-Andersen et al. (2017). For a comprehensive review refer to (Tautz-Weinert and Watson (2016))).

SCADA data based condition monitoring therefore represents a cost efficient and effective complement to state-of-the art CM-solutions. Its primary task is to classify the state of a turbine or one of its components as either healthy or faulty. However, the available SCADA data represents predominantly healthy operation with no or only comparatively few instances of faulty condition. In such a setting semi-supervised anomaly detection, often called normal behaviour modeling, has proven to be useful (Chandola et al. (2009)). Normal behaviour models (NBMs) are trained on healthy turbine data to represent the class

corresponding to normal state. Deviations between model output and the measured SCADA sensor values can be processed and evaluated to identify anomalies (compare Figure 1). Zaher et al. (2009) were among the first to apply the approach in the wind domain and prove its feasibility. Many publications with successful early detection of malfunctions followed (compare e.g. Butler et al. (2013), Kusiak and Verma (2012), Sun et al. (2016), Bangalore et al. (2017) and Bach-Andersen et al. (2017).

Despite the promising NBM examples reported in literature scaling the method to large fleets of wind turbines comes with

practical challenges. Leahy et al. (2019) analysed 12 studies that apply the concept of NBM to wind turbine SCADA data and found that all but one reported significant manual efforts in data pre-processing due to data quality and data access related issues. That is why researchers have developed different filtering methods with the aim to ensure healthy training data without traces of malfunctions. They can be divided into domain-knowledge-based-, alarm-based-, work-order-based-, or statistical-approaches (Leahy et al. (2019)). Manual selection of representative operational patterns from the SCADA data sets would

be an example of domain-knowledge-based filtering and can be found for instance in Zaher et al. (2009). Another common procedure is to filter NBM data against a certain threshold of active power production in order to exclude transitions between operational and non-operational states as well as corrupted sensor measurements during standstill (compare e.g. Sun et al. (2016) ,Bangalore et al. (2017), Tautz-Weinert (2018)). Schlechtingen and Santos (2011) were among the first to describe a more systematic semi-automated data pre-processing procedure. It consists of a domain-knowledge-based parameter range





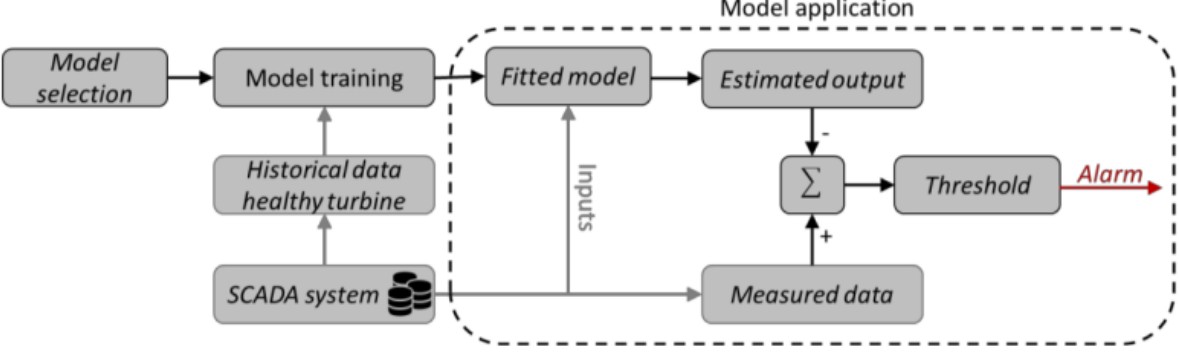

**Figure 1.** Scheme of normal behavior model-based anomaly detection with offline model preparation (left) and online application (right).

check, data scaling, handling of missing values and lag removal. These measures have been extended by multivariate statistical filter methods to automatically remove outliers (compare e.g. Bangalore et al. (2017)). However, a much more severe problem than missing, invalid or poorly processed data is caused by structural changes in sensor measurements (Tautz-Weinert and Watson (2017)). Abrupt changes in the underlying data generating regime at a specific point in time (change-point) have

been reported in different publications (e.g. Schlechtingen and Santos (2011) or Tautz-Weinert and Watson (2017)). They can be caused by sensor or component malfunctions as well as by maintenance actions. In an ideal setting, all potential causes would be detected, corrected, fully documented and available to the respective data analyst. Unfortunately, this is rarely the case in practice (Tautz-Weinert and Watson (2017) and Leahy et al. (2019)) which has severe implications for NBMs. NBM training represents statistical parameter estimation of an underlying process which can only be successful if training data

is stochastically homogeneous. NBMs trained on data containing change-points (CPs) are fit to multiple, potentially even abnormal, states of operation causing them to fail their intended task. Since CPs can make the NBM-approach infeasible in practice, this has been identified as the most serious issue for their application (Tautz-Weinert and Watson (2017)). Based on the above findings this study aims to be the first one to conduct a systematic analysis regarding the presence of CPs in SCADA signals. Moreover, an approach for robust detection of structural changes in SCADA measurements will be suggested.

Non-parametric kernel-based change-point detection (CPD) methods will be adapted to the problem at hand. This includes recommendations for the choice of respective hyperparameters and useful signal pre-processing steps based on evaluation across a large range of SCADA signals from muliple wind farms. The result represents a step towards scalability of SCADA based NBM which is essential for the promising method to reach its full potential. The remainder of this paper is organised as follows: Section 2 gives an overview of CPs presence in the SCADA database with summary statistics and characteristic

examples. Section 3 presents the method utilised in this study by formalising the CPD problem, introducing kernel-based CPD algorithms and their respective evaluation metrics. Section 4 specifiesthe CPD algorithm with its pre-processing steps and the selection of hyperparameters. Section 5 presents the performance over a range of hyperparameter configurations with respect to different evaluation objectives followed by a discussion of results. Section 6 concludes with a summary and outlook.



## 2 Structural breaks in SCADA data measurements

Wind turbine SCADA systems record measurements from sensors placed all over the turbine. Available signals usually include temperature measurements, electrical measures, pressure values, speed counters, timers, status parameters and environmental conditions. Modern SCADA systems often record more than 100 different signals. The typical temporal resolution is 10 minute average values, although systems with a resolution as high as 1 Hz exist but are far less common. Additionally, some manufacturers store the signals' standard deviation as well as minimum and maximum values during the averaging period. Structural breaks in these measurements manifest themselves as an abrupt change in sensor behaviour at a specific time instant $\tau$ called a CP. The various potential causes can be classified into being sensor, component or maintenance related. Sensor related structural breaks can be caused by sensor drifts, sensor failures or malfunctions in the communication system. Component related breakpoints can originate from particularly strong wear or component failure. Lastly, maintenance induced changes can be attributed to specific actions like the exchange of operating materials, replacement of components or sub-components and control adjustment. Although, the presence of CPs has been described in multiple publication as a challenge when working with wind turbine SCADA data (compare Tautz-Weinert and Watson (2017) and Leahy et al. (2019)), this study is the first to systematically evaluate the presence of CPs in wind turbine SCADA signals, to the author's knowledge.

### 2.1 Data base and change-point annotation

For the current study SCADA data from 33 multi-MW turbines from 3 different sites was used. For each turbine SCADA data representing more than 2 full years of continuous operation was present. Each turbine's SCADA system records between 30 and 100 signals. From the almost 2000 time series 600 were selected for CPD based on the signal's potential for NBM. Therefore, all power train related temperature and pressure values were selected. Additionally, temperatures from the pitch system, the electrical system, and ambient conditions were chosen. The left pie-chart in Figure 2 shows the allocation of the 600 analysed signals to the respective components. Next to the self explanatory component classes the category 'Others' contains signals such as shaft bearing, nacelle and brake temperatures. Generator and gearbox-related signals represent half of the overall selection. These are also the components typically targeted by SCADA based NBMs (compare Tautz-Weinert and Watson (2016)). The high number of pitch-related signals is due to the availability of temperature measurements from multiple sub-components on each blade's pitch systems. A full list of the analysed signals and their mapping to the respective components can be found in Appendix B1. In addition to the sensor data SCADA-log files, information about major maintenance activities was present. This information was combined with a visual inspection of all analysed time series to manually annotate CPs. The raw signals, their de-trended and normalised transformations (compare section 4.1) as well as their summary statistics (minimum, mean, median and maximum) were compared with different temporal resolutions. Additionally, the time series were contextualised by comparing all signals related to the same component. Often changes in one component were reflected in multiple component related signals at the same point in time. Such coherent findings helped to increase confidence during the annotation process. Additionally, each signal was compared to its equivalent from at least 5 neighbouring turbines in the farm. This so called trending approach is well known in SCADA analysis for monitoring wind turbines (compare Tautz-Weinert and



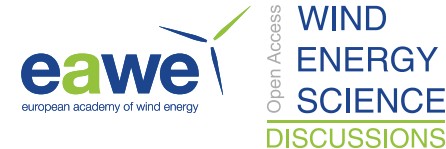

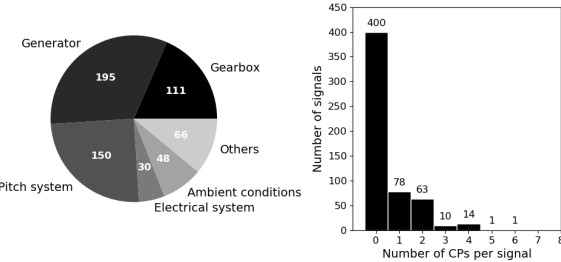

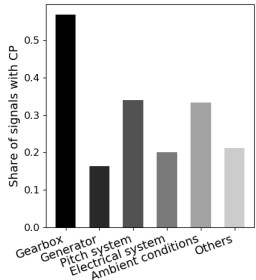

**Figure 2.** Number of signals per component (left), number of CPs per signal (center), and share of signals with CPs per component (right) for the full 2-years time horizon.

Watson (2016)) and helped to highlight the difference between normal signal behaviour and abrupt changes. Moreover, after annotating each signal for the full two years, smaller segments were analysed to judge the signal's behaviour in a different temporal context. The result of this tedious task was reviewed by fellow researchers in order to secure utmost objectivity. Nevertheless, not every annotated CP could be confirmed with the maintenance information or the SCADA logs which is

attributed to the incomplete available information, a typical problem for this field of research (compare e.g. Tautz-Weinert and Watson (2017) and Leahy et al. (2019)). However, the described signal inspection procedure is assumed to reduce the number of mis-annotations to a minimum.

Figure 2 shows the results of the signal annotation process. The central chart represents a histogram over the number of CPs per signal. Exactly one third of the analysed signals were affected by changes over the approximately 2.5 year period.

However, generally only a few CPs were found per signal. Actually, less than 5 % of the affected signals exhibit 3 or more CPs. The right-hand diagram of Figure 2 compares the share of signals corrupted by changes for each component category. Gearbox related signals are most affected with more than half the signals containing CPs. For pitch related and ambient condition signals around one 30% of the time series were found to be affected. The high number of pitch related CPs were caused by systematic disturbances in the pitch motor temperature sensors for one of the wind farms. In case of ambient conditions a

range of temperature sensors was found to be affected by severe drifts. The extent of CP presence highlights the necessity of a robust CPD methodology. The presented figures reflect the CP summary statistics across the selected signals for the full available period. In addition the 600 signals will be split into 1200 signals (covering approximately 1 year each) to analyse the algorithms ability to generalise to different signal lengths. This obviously changes the summary statistics which can be found in Appendix A1.

## 2.2 Signal and change-point characterisation

Changes manifest themselves in a wide range of different signal behaviours, due to the multitude of potential reasons for structural changes, as well as the unique statistical natures of each signal. This is why a unifying framework to detect changes in SCADA measurements has to account for the diversity of signals and changes. Generally CPs can be classified into being





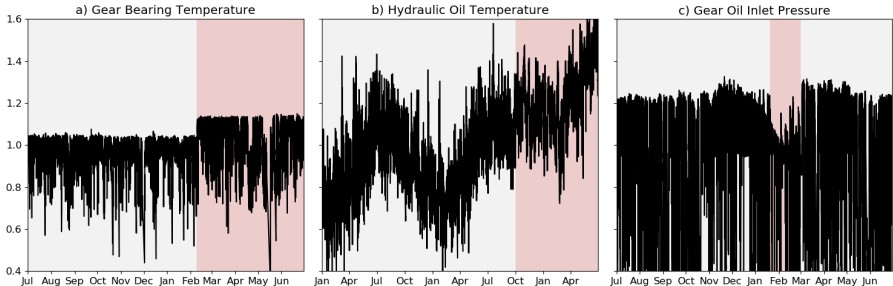

**Figure 3.** Exemplary SCADA signals exposing different structural changes. CPs are indicated by a change in background color.

abrupt or gradual as well as permanent or temporary. Figure 3 shows three inherently different SCADA signals that exemplify different types of structural changes. In order to highlight the changes non-operational data was excluded and the signals were normalised with their respective median to facilitate a comparison. Figure 3 a) shows a gearbox bearing temperature. Since the gearbox is an actively cooled component the signal has a comparably small, static range and a well defined maximum.

Therefore, the CP in February of the depicted year is easy to recognise. It occurred after a scheduled maintenance during which a cooling fluid was exchanged and the bearing consequently operates at clearly elevated temperatures. Such abrupt changes are mostly induced by maintenance actions or spontaneous sensor and component failures. Their detection and especially their exact localisation is usually easy in comparison to gradual changes due to their abruptness. Figure 3 b) displays a turbine's hydraulic oil temperature over a period of two years. The signal is characterised by a comparatively large range and significant

seasonality. A hydraulic fault in October of the second depicted year of operation causes the temperature to steadily rise compared to pre-CP conditions. The overlapping effects of seasonality and the gradual nature of the underlying change make the determination of the CP as well as its exact point in time challenging. This holds true even if additional information is available. A reported finding in a maintenance protocol, for instance, confirms the presence of a structural change but represents only an upper bound on the actual time of the event. Moreover, normal wear of components can also induce a steady gradual

change and it can be difficult to tell normal wear apart from additional gradual effects. Here, the comparison of signals from neighboring turbines has proven to be helpful. Lastly, Figure 3 c) shows a gear oil pressure signal over a period of one year. The signal with high variance and a relatively well defined maximum shows a temporary decline of a turbine's gear oil inlet pressure and its return to the initial level. This was caused by an issue with the lubrication oil filter which was fixed during a scheduled maintenance activity. Most temporary changes are caused by corrective maintenance action related to an initial change. When

formalising the CP detection problem in the next chapter it becomes clear that the individual signal characteristics and their change in behaviour translate into how statistically distinct and therefore, how easy to detect, a CP is. This depends on the ratio between the magnitude of change and the individual signal variance or noise level (Garreau (2017)), but also to the duration of how long a certain change is present. Intuitively, temporary changes or CPs at the very beginning or end of a time series are therefore more difficult to detect.



## 3 Method for change-point detection

The detection of CPs in time series is a well studied problem in statistics, signal processing and machine learning. The goal is to detect time instants at which the underlying data generation process and therefore the marginal distribution of the observations changes abruptly. In other words the time series is to be split into statistically homogeneous segments (Brodsky and Darkhovsky

(1993)). First works date back to the 1950s (e.g. Page (1955)) but the topic has stayed the subject of active research until today, with methods being further refined and applied to many different domains, such as remote sensing (Touati et al. (2019)), audio signal processing (Rybach et al. (2009)), or medical condition monitoring (Malladi et al. (2013)). Refer to Aminikhanghahi and Cook (2017) for an overview on time series CPD methods. The following section will describe, classify and formalise the CP problem at hand based on Brodsky and Darkhovsky (1993).

### 10 3.1 Problem formulation

Conceptually, the CPD problem can be divided into online and offline detection. The former, sometimes also referred to as sequential CP detection, aims to identify changes in real-time settings as early and confidently as possible. In contrast the latter, also known as signal segmentation, aims to determine the CP a posteriori with the data acquisition process being completed at the time that the homogeneity hypothesis is checked. Offline CP-problems can be further classified with respect

to the a priori knowledge of the respective task. Complexity is significantly lower if the number of true CPs is known which reduces the task to the precise estimation of their location. In most real-world applications, however, the number of CPs itself has to be estimated. The same applies for a priori information about the statistical characteristics of the respective signals. Prior knowledge allows for assumptions regarding the family of underlying distributions. Therefore, CPs can be detected by identifying a change in the parameters describing the distribution. Non-parametric methods on the other hand require no such

prior information which makes them more flexible and therefore often better suited for real world problems. The present task of ensuring CP-free training data sets represents an offline CPD problem, where the number of true CPs is unknown. Even though it is expected that many SCADA signals are not affected by structural changes it is possible that more than one statistically homogeneous segment exist per signal. Lastly, the SCADA data set consists of various statistically different signals which do not allow for unifying assumptions regarding their family of distributions. Therefore, non-parametric methods will be applied.

Let's formalise the given problem under the prevailing conditions. We assume $X = \{X_1, X_2, ..., X_T\}$ to be a piece-wise stationary time series signal in $\mathbb{R}^d$ consisting of $T$ observations. Piece-wise stationarity implies that $X$ can be divided into $N$ ($N \geq 1$) segments where each segment is well described by some distribution which might differ for consecutive segments. The segments therefore represent homogeneous sets $s$ which are characterised by $N-1$ CPs at some unknown instants in time $\tau_1^* < \tau_2^* < ... < \tau_{N-1}^*$ (1). Now, CP detection can be formulated as a model selection problem where the CPs $\tau$ are the model

parameters to be estimated. This can be achieved by defining a cost function $C(\tau)$ that quantifies intra-segment dissimilarity with respect to the true CPs $\tau^*$ (2). A naive minimisation of this cost function would result in a segmentation into N segments





of unit size. Therefore, a regularisation term was proposed for example by Lavielle (2005) which penalises for every additional CP and therefore reduces complexity of the segmentation (2).

$$s = \{s_1, s_2, ..., s_N\} = \{\{X_0, ..., X_{\tau_1}\}, \{X_{\tau_1+1}, ..., X_{\tau_2}\}, ...., \{X_{\tau_{N-1}}, ..., X_T\}\} \tag{1}$$

$$\hat{\tau} \; \epsilon \; \underset{\tau}{\arg\min} \; C(\tau) + \mathcal{P}(\tau) \quad \text{where} \quad C(\tau) = \sum_{n=1}^{N} C(s_n) \tag{2}$$

Since the complexity of the optimisation problem grows quadratic with the number of data points a naive approach for minimising the cost function $C(\tau)$ can be computationally expensive. Several approximate search methods like a sliding window or binary segmentation were developed (compare Truong et al. (2020)). They come with benefits regarding computing time but naturally compromise on precision. The optimal solution can still be obtained efficiently by applying an algorithm based on dynamic programming. It was originally introduced in 1958 (Bellman (1958)) for solving a shortest-path problem for

traffic networks. Since then the algorithm has been developed further (see e.g. Guédon (2013)) and was successfully applied in the context of CPD. The method utilises the additive structure of the cost objective to recursively compute optimal CPs for multiple sub-signals among which the global minimum is then selected. An implementation of the algorithm is publicly available as part of the CP detection library `ruptures` in Python (Truong et al. (2020)) and was utilised within this study.

### 3.2   Kernel based change-point detection

Equation 2 represents a general cost-function for solving the signal segmentation task at hand but the result heavily depends on an appropriate measure for the intra-segment similarity. Harchaoui and Cappé (2007) proposed a kernel-based approach which does not rely on parametric assumptions but is able to detect changes in the high order moments of the signal distribution. Kernel methods use mapping functions $\Phi : \mathbb{R}^d \to \mathcal{H}$ to implicitly project a signal into a potentially much higher dimensional Reproducing Kernel Hilbert Space (Scholkopf and Smola (2002)). With the well known kernel-trick the distance or similarity

of two data points in the high dimensional feature space can be calculated by directly applying the kernel function (compare Eq. (3)). Harchaoui and Cappé (2007) used this property to evaluate the adequacy of $\tau$. They define a kernel least-squares criterion that measures the intra-segment scatter (see Eq. 4). Intuitively, the second term of Equation 4 increases if the chosen segments are more similar to each other and in return maximises dissimilarity between segments due to the negative sign. Note that the intra-segment scatter requires the calculaten of the kernel-gram matrix $G_{i,j} = K(X_i, X_j)$, which implies a quadratic

computational complexity and therefore restricts the method regarding the size of the data sets. By minimising the criterion the best segmentation for a known number of CPs can be obtained. Conceptually, any positive semi-definite kernel can be applied in this framework. Popular candidates are the Linear ($i$), Laplacian ($ii$) or Gaussian ($iii$) kernel (compare Eq. 5). Note, that Laplacian and Gaussian kernels need the selection of an appropriate bandwidth parameter $h$. Arlot et al. (2016) expanded the method to an unknown number of CPs by applying the concept of penalising for additional CPs (compare Eq. 2). Since then



the kernel based algorithm has been successfully applied to multiple real-world time series CPD problems (compare Arlot et al. (2016)).

$$k(x, x') = \langle \Phi(x), \Phi(x') \rangle_{\mathcal{H}} \qquad \text{and} \qquad k(x, x) = \|\Phi(x)\|_{\mathcal{H}}^2 \tag{3}$$

$$C(\tau) = \frac{1}{T} \sum_{t=1}^{T} k(X_t, X_t) - \frac{1}{T} \sum_{n=1}^{N} \left[ \frac{1}{\tau_n - \tau_{n-1}} \sum_{i=\tau_{n-1}+1}^{\tau_n} \sum_{j=\tau_{n-1}+1}^{\tau_n} k(X_i, X_j) \right], \tau_0 = 0 \tag{4}$$

$\quad i) \; k_{lin}(x,y) = \langle x, y \rangle \qquad ii) \; k_{lp}(x,y) = \exp\left(\frac{-\|x-y\|}{h}\right) \qquad iii) \; k_{rbf}(x,y) = \exp\left(\frac{-\|x-y\|^2}{h}\right) \tag{5}$

### 3.3 Performance evaluation

The performance of the CPD algorithm can be evaluated using the classic notation of true positives (tp), false positives (fp), true negatives (tn) and false negatives (fn). In order to appropriately interpret the evaluation results the implications of false classifications have to be considered. In case of NBMs a fn translates into a risk for model quality, a fp into loss of potentially

valuable training data. However, the individual impact depends on the severity of the change, meaning the more distinct from normal signal behaviour as well as the longer its presence, the more severe its consequences. This goes well with the concept of the presented CPD algorithm since the notion of severity directly translates into a cost reduction by segmentation. For the concrete evaluation of CPD results two different evaluation objectives are distinguished:

1. Automatic training data validation: detect presence of CP in given processed signal

2. Automatic training sequence selection: detect number and exact locations of CPs in processed signal

Automatic training data validation answers whether a CP is present in a given SCADA signal or not. Therefore, the CP-detection result is evaluated once for each signal. In case the algorithm indicates one or more CPs for a signal containing at least one CP the result is evaluated as a true positive. No CP detection in a CP-free signal represents a true negative, etc. In practice this means that a CP detection result is evaluated as a true positive, even if the number and location of indicated CPs

does not necessarily represent the ground truth. Nevertheless, this can be a useful information for validating signals against the presence of CPs. Especially, since the alternative is a full manual inspection of all signals. Automatic training data validation therefore pre-selects the signals for visual inspection, in which the actual locations of the CPs are subsequently determined. The next step towards a fully automated NBM approach is auotmated training sequence selection. This requires a more precise evaluation for each CP and each CP-indication individually. Therefore, an acceptable margin is selected around each true CP

in which a detection is evaluated as a tp. While for automatic training data validation this margin was practically set to infinity a fixed number of days has to be chosen for automatic training sequence selection. A CP which is present in the signal but not indicated by the algorithm is evaluated as a fn. Detection outside the margin boundaries represents a fp. A tn represents a





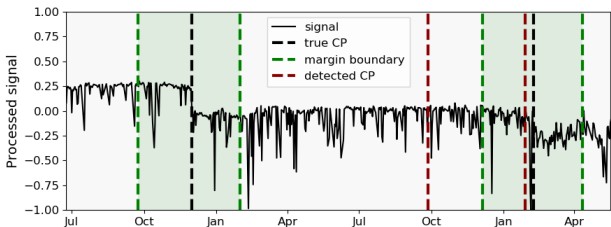

**Figure 4.** Exemplary evaluation of a CP-detection result with one fn (left), fp (center) and tp (right).

CP-free signal with no detection. The concept is visualised in Figure 4 where a tp, fp and a fn are depicted. Note, that a CP indicated just outside the margin already leads to a "double" punishment by evaluating the indication as a fp and the true CP as not detected (fn). Intuitively, the overall detection result depends on the selected acceptable margin. Moreover, the margin corresponds to the amount of data around the detected CP to be automatically cut off in automated training data selection and

5 therefore to a trade-off between data loss and accuracy. In this paper the acceptable margin was selected to be +/- 60 days around the true CP. The choice is motivated by the fact that missing a true CP can be much more critical for many applications, such as NBM for example, than a the reduction of training data by the given period. Moreover, this attributes to potential inaccuracy during the manual annotation process due to uncertainties about the actual occurance of the change (compare section 2.1). With the classification of each detected CP into true or false positive or negetive, the well known evaluation metrics Accuracy,

Precision, Recall and the F1-Score can be calculated:

$$accuracy = \frac{\#tp + \#tn}{N} \qquad precision = \frac{\#tp}{\#tp + \#fp} \qquad recall = \frac{\#tp}{\#tp + \#fn} \tag{6}$$

$$f1 - score = 2 * \frac{precision * recall}{precision + recall} \tag{7}$$

## 4  Algorithm for change-point detection

This section describes the detailed steps of signal processing applied to detect the CPs in this study. Signal-pre-processing as

well as the choice of hyper-parameters are discussed.

### 4.1  Data pre-processing

Wind turbine operation is highly volatile due to intermittent ambient conditions. This is reflected in the high variance of raw SCADA measurements and complicates CP detection because the change in signal behaviour might be small in relation to regular signal behaviour. Signal pre-processing methods can help to reduce the signal to its most valuable components for CP





detection and therefore facilitate the process. Tautz-Weinert and Watson (2017) suggest the comparison of monthly maximums and percentiles to detect structural changes. This was found to be too granular to attribute for temporary changes of less than one month. Additionally, such an approach does not reduce seasonality which was found to be an important factor for successful kernel-CPD. Instead, the following pre-processing steps were taken in this study:

- Removal of non-operational periods

- Normalisation with operational state and ambient conditions

- Re-sampling with reduced temporal resolution

The removal of non-operational periods is a routine pre-processing step in SCADA data analysis (compare e.g.Sun et al. (2016) ,Bangalore et al. (2017), Tautz-Weinert (2018)) and was motivated by the reasoning that changes in operational con-
ditions will become most apparent when the turbine is in operation. Also, it was observed that sensor values during non-operational periods, e.g. during maintenance, sometimes take pre-defined standard values. In order to exclude such distorting effects all data points where the turbine is operating on less than 10% of its rated power were excluded. Furthermore, the signal measurements were normalised with respect to the prevailing operating conditions. Active power production and rotor rotational speed were found to be the most dominant to characterise the turbine's operational state. Ambient temperature was
identified to be a good regressor to exclude seasonality from the sensor measurements. Therefore, each signal was normalised using these three input variables in a linear regression (compare Eq. 8). The model was found to adequately subtract the influences of external conditions, is computationally cheap and due to its simplicity does not allow overfitting the sensor signals. In a last step, the normalised signal was averaged over each day, if at least three hours of operational data were available. This allows to extract the normalised signal characteristics and additionally reduces the amount of data points which facilitates the
computation of the kernel gram-matrix. An exemplary result of the pre-processing procedure is shown in Figure 5. It displays the three signals shown earlier (compare Figure 3), after pre-processing. Note in particular how the method facilitates a clear identification of the CP in the hydraulic oil temperature compared to the raw signal.

$$signal^* = f(X_P, X_{rpm}, X_{T_{amb}}) = w_1 \cdot X_P + w_2 \cdot X_{rpm} + w_3 \cdot X_{T_{amb}} + c \qquad (8)$$

A minor disadvantage of this approach is that the regressing signals, meaning active power, rotor speed and ambient temper-
ature cannot be pre-processed the same way. In fact, abrupt change in one of the regressors can induce a CP in highly correlated signals. In this study such a case occurred a few times when the ambient temperature sensor of a turbine was corrupted. However, by regressing the inputs themselves with signals from neighbouring turbines first and then running the algorithm on them to check for CPs can exclude those cases. Alternatively, a simple rule checking for simultaneous CP detections in all signals particularly correlated to the same regressor can do the trick as well.



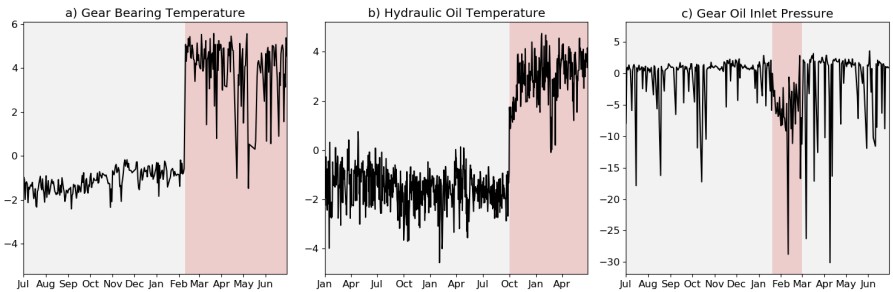

**Figure 5.** Exemplary SCADA signals after pre-processing (compare Figure 3). True CPs are indicated by a change in background color.

## 4.2 Choice of hyperparameters

The CPD method described in section 3 requires adequate selection of several hyperparameters. Namely, the type of kernel, its respective bandwidth and the penalty term for additional CPs. All three were found to have profound impact on the CPD performance. Choosing an appropriate kernel is a well studied problem by itself in many applications. In the context of CPD

the widely used Linear, Gaussian and Laplacian kernels have been used (Garreau (2017)). Therefore, all three will be compared within this study. The choice of an adequate bandwidth $h$ is another problem often encountered when working with kernel methods. Looking at the definition of the Gaussian and the Laplacian kernels (see Eq. (5)) it becomes clear that a bandwidth chosen too large or too small will make the entries of the gram matrix go towards zero or one respectively and therefore valuable information will be lost. A common approach is therefore to choose the bandwidth in the range of the calculated distances. Gretton

et al. (2012) for example suggest a median heuristic in the context of a kernel two-sample test (compare Equation 9). This heuristic is heavily used in ML literature (Garreau (2017)) and is also applied in the CPD settings (Truong et al. (2020)). Arlot et al. (2016) on the other hand suggest to use the empirical standard deviation of the signal itself as the bandwidth. Both choices of bandwidths are tested and compared in this paper. Furthermore, it is argued here that estimation of an appropriate bandwidth based on a signal with abruptly changing properties might lead to a non-optimal choice. Therefore, a third approach is being

introduced and tested where the signal is divided into $k$ different segments $S_{bw} = \{\{X_0...X_t\},...,\{X_{(k-1)*t+1}...X_{k*t}\}\}$ of equal length $t$ and the empirical standard deviation is calculated for each segment. The bandwidth is consequently chosen as the maximum of the $i$ standard deviations (compare Equation 9). In this study $k$ is selected to be 20. Consequently each segment consists of roughly 2 months of operational data. For the remainder of the paper aproach c) is refered to as batch-std bandwidth.

$a)$    $h = median(\|X_i - X_j\|^n)$      $b)$    $h = std(X)$      $c)$    $h = max(std(S_{bw}))$          (9)

Another crucial hyperparameter choice is the selection of an appropriate penalty term (compare Equation 2) which controls the number of CPs to be detected by the algorithm. If the penalty is selected too low, too many CPs will be detected and vice



versa. A data driven approach for choosing the penalty in context of minimisation of a penalised creterion is the so-called slope-heuristic (Birgé and Massart (2007). It was shown that the optimal penalty to avoid overfitting is approximately proportional to a minimal penalty which can be obtained based on a regression between the penalised quantity and the associated cost function without penalisation. In the context of CP detection this was firstly applied by Lebarbier (2002) and further refined by Baudry

et al. (2012). They suggest a minimal penalty based on two constants $s_1$ and $s_2$ which are obtained by a regression between the cost function $C(\tau)$ and $\log\left(\binom{T-1}{D_\tau-1}\right)$ as well as $\frac{D}{T}$ for $D \in [0.6 \cdot D_{max}, D_{max}]$ (compare 10). Finally, the minimal penalty is multiplied with the factor $\alpha$ to obtain the final optimal penalty. Even though the optimal choice of $\alpha$ is problem specific $\alpha = 2$ was reported as a suitable choice Arlot et al. (2016).

$$pen_{opt-slope}(\tau) = \alpha_{slope} \cdot \frac{1}{T} \cdot \left(-s_1 \cdot \log\binom{T-1}{D_\tau-1} - s_2 D_\tau\right) \tag{10}$$

In this study, the slope heuristic is compared to a simpler approach chosen based on the following consideration: signals which are inherently similar to themselves are by default characterised by a relatively low initial cost value and vice versa. This means that each CP by default leads to a larger cost reduction for more dissimilar signals. Therefore, the penalty term is chosen based to the the sum of costs without any CP. Figure 6 supports this reasoning. Here, a CP was enforced on all signals without changes. The resulting reduction in the cost function is shown over the initial average cost (left). An approximately

quadratic relation between cost reduction and initial average cost can be observed. The right side of Figure 6 shows the relative reduction normalised with $C(\tau = 0)^2$. Consequently, the normalised cost reductions are much more uniformly distributed and facilitate the selection of a single penalty value over all signals. Moreover, the penalty term can now be easily calculated form the signal characteristic itself. This is considered an advantage over the more complex methods found in literature. The findings indicate that a reasonable choice of the penalty factor $\alpha_{cost}$ would be in the range between 75 and 150 for a Laplace kernel with

a bandwidth selected according to the batch-std heuristic. A penalty factor larger than 200 can be considered a conservative choice with only a few false positives. Reversely the reduction induced by a CP can be interpreted as a confidence measure. Note that these values depend on the kernel configuration.

$$pen_{opt-cost}(\tau) = \alpha_{cost} \cdot C(\tau = 0)^2 \tag{11}$$

## 5   Results for change-point detection

The result section presents the algorithm's performance on automatic training data validation and selection. For both evaluation objectives different hyperparameter configurations in terms of kernel-, bandwidth- and penalty-selection are compared. Additionally, the effect of signal length is investigated in order to ensure the algorithms generalisation abilities. Results are analysed on a cumulative as well as on a component level. Finally, the results are discussed and implications for the algorithm's practical application derived.





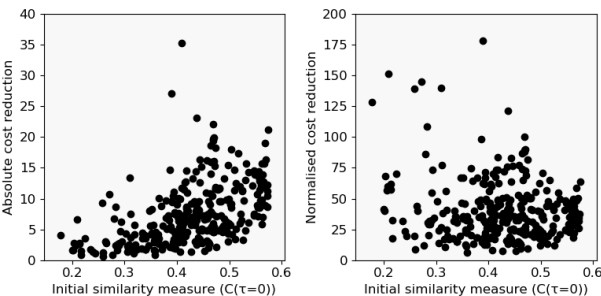

**Figure 6.** Absolute (left) and normalised (right) cost reduction of healthy signals by imposing one CP versus average cost.

## 5.1 Results automatic training data validation

In this section the algorithm's ability to distinguish between signals with and without CPs is evaluated. Figure 7 shows the results achieved by different configurations on the full signal length (left) and the half signal length (right), representing approximately 2 and 1 years of operation. Both, F1- and accuracy-scores are compared for different kernels, bandwidth choices and penalty selection schemes. A clear ranking can be identified when comparing kernels. For configurations with cost based penalties Laplacian kernels perform best followed by Gaussian kernels. Linear kernels perform much worse. For penalties chosen according to the the slope-heuristic the contrary is the case. Linear kernels perform best, closely followed by Laplace and Gaussian configurations. In terms of bandwidth selection, the intra-kernel ranking differs, but the leading Laplacian configurations use the batch-std heuristic. It clearly outperforms established standard deviation or median heuristics. When comparing penalty selection schemes the cost-based penalty estimation suggested in this paper clearly outperforms the slope-heuristic. All discussed qualitative observations hold for both time horizons. However, a clear performance loss in terms of F1-score for the shorter signals can be observed. This is attributed to the design of the evaluation scheme. Firstly, there is a shift in the distribution between affected and not affected signals (compare Figures 2 and A1. Secondly, in case of multiple CPs it is enough to flag the most significant one to be classified as a true positive in the two year signal, whereas splitting the might require detection of a less severe change in one half of the signals.

In absolute terms the overall best performing configurations are able to classify more than 90% of the signals correctly regarding the presence or absence of CPs. The wrongly classified signals are approximately one quarter false positives and three quarters false negatives. This translates into F1-scores of 0.87 and 0.76 for the different signal lengths. This performance is reached for a penalty factor of $\alpha_{cost_2} = 145$ and $\alpha_{cost_1} = 80$ respectively. The best result using the slope-heuristic for penalty selection was achieved for penalty factors of $\alpha_{slope} = 12.5$ which is much higher than the $\alpha_{slope} = 2$ suggested in literature. At the same time F1-scores range around 20% behind the leading cost-penalty based configuration. Table 1 displays the overall results as well as the results per component for the best performing CPD configuration on the 2 year signals in detail. The algorithm reaches high performance across components. Ambient condition signals as well as gearbox and pitch related signals were classified with particularly high accuracy. Correctly classified CPs are often characterised by sharp transitions between





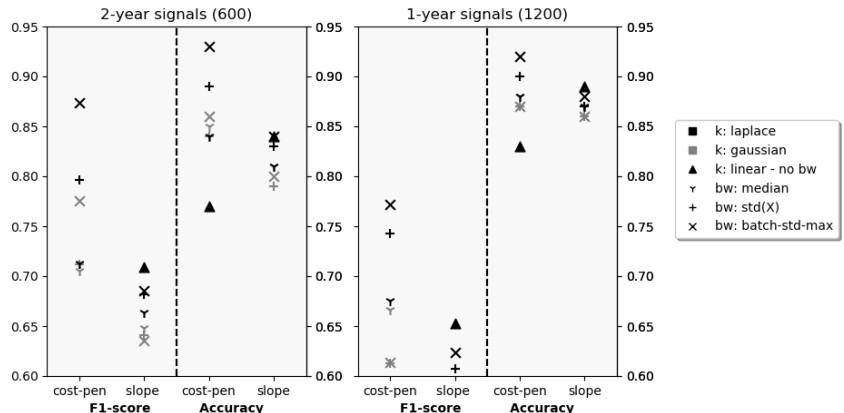

**Figure 7.** Validation F1-score and accuracy for different hyperparamter configurations on 2 year and 1 year signals

**Table 1.** Automated signal validation results per component for best configuration (Laplace / batch-std / $\alpha_{cost} = 145$) on 2-year-signals

| Component | tn | tp | fn | fp | accuracy | precision | recall | f1-score |
|---|---|---|---|---|---|---|---|---|
| Gearbox | 40 | 57 | 6 | 8 | 0.874 | 0.877 | 0.905 | 0.891 |
| Generator | 162 | 21 | 11 | 1 | 0.938 | 0.955 | 0.656 | 0.778 |
| Pitch system | 98 | 38 | 13 | 1 | 0.907 | 0.974 | 0.745 | 0.844 |
| Electrical system | 24 | 4 | 2 | 0 | 0.933 | 1.0 | 0.667 | 0.8 |
| Ambient conditions | 30 | 16 | 0 | 0 | 1.0 | 1.0 | 1.0 | 1.0 |
| Others | 50 | 11 | 3 | 2 | 0.924 | 0.846 | 0.786 | 0.815 |
| Total | 404 | 147 | 35 | 12 | 0.91 | 0.925 | 0.808 | 0.862 |

the states and a significant difference in relation to the regular signal noise. Examples for successful detections can be found in Figure 8. The top chart shows the correctly classified change in gear oil pressure after maintenance. The second chart shows the correctly identified shift and its return to normal behaviour in a nacelle temperature signal which was induced by problems in the generator cooling system and its consecutive fix. At the same time, there is a relatively high number of false positives for gearbox related signals. This is mostly caused by bearing temperatures that are gradually rising due to normal wear (compare Figure 9 $a$)). The algorithm detects the drift in the signal's distribution which, under the given evaluation framework, represents a false positive. In the broader context of NBM this information is still valuable since it highlights the need for periodical model re-training. False negatives were mostly caused by short temporary changes which were not pronounced enough to compensate for the penalty of two CPs, which would be required to flag them correctly. An example is shown in Figure 9 $b$) which depicts a generator bearing temperature with temporary high temperatures. The detailed results for all configuration by penalty and time horizon can be found in Appendix C.





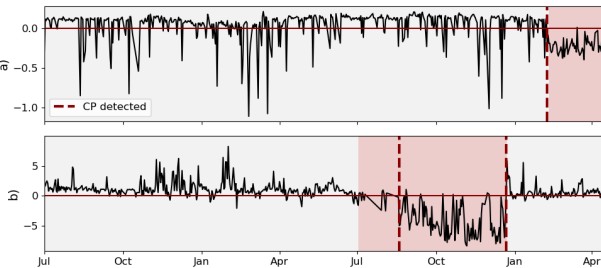

**Figure 8.** Examples of successful CP detection in gear oil pressure (a) and nacelle temperature (b). Change in background colour indicates real CP, dashed lines detections.

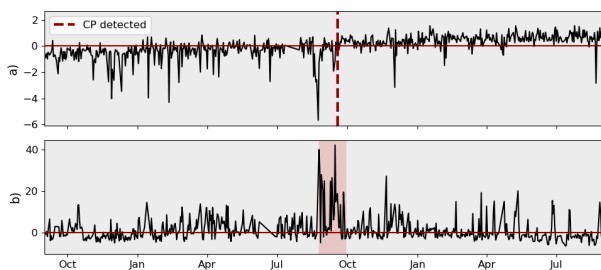

**Figure 9.** Examples of misclassification in gearbox (a) and generator (b) bearing temperature. Change in background color indicates true CPs, dashed lines detection.

## 5.2 Evaluation of automated training sequence selection

In this section the algorithm's ability to automatically select periods without CPs for each signal is analysed. Therefore, performance is evaluated for each CP individually rather than for each signal (compare section 3.3). Figure 10 shows the CPD results analogously to the results for automated training validation in the previous section. Qualitatively, the findings with respect to kernel selection and configuration are equivalent. Laplace kernels with batch-std bandwidths perform best. In comparison with the results from the previous section the more difficult evaluation objective manifests itself in overall lower performance scores. Even though accuracies reach well above 80%, F1-scores drop to 0.73 and 0.71 for the two time horizons. However, the performance between the two analysed time horizons are very similar which attributes for the algorithm's ability to generalise across different signal length. The optimal penalty factors remain time horizon specific but stable across evaluation metric with $\alpha_{cost_2} = 150$ and $\alpha_{cost_1} = 80$. The general advantage of cost based penalties is preserved with F1-scores approximately 15% above the best slope-heuristics results which are achieved at $\alpha_{slope_2} = 11.5$ and $\alpha_{slope_1} = 4$.

Table 2 displays the overall results as well as the results per component for the best performing CPD configuration on the 2 year signals in detail. To explain the drop in F1-score the different false classification of each component were analysed. Gearboxes show both, a relatively high number of fns as well as fps. Approximately 50% of the fns can be attributed to the





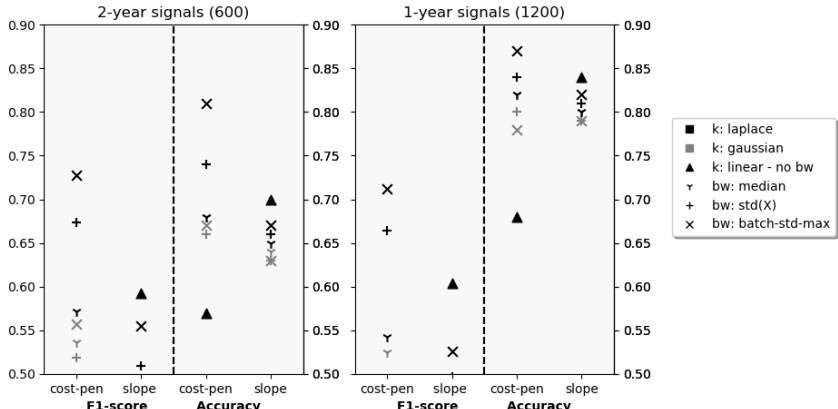

**Figure 10.** Selection F1-score and accuracy for different hyperparamter configurations on 2 year and 1 year signals

coexistence of large and comparatively small changes in the same signal. An example is shown in Figure 11 $a$) where an oil temperature signal undergoes two significant changes with the second one not being detected. The initial dissimilarity, based on which the penalty is calculated, is dominated by the first change and therefore detection of the second change cannot compensate for the high penalty value. Another 20% of the gearbox-related fns are caused by short temporary changes and

further 20% by detections outside the 60 days margin (compare e.g. the gearbox bearing temperature in Figure 11 $b$)). These represent at the same time approximately 25% of the fps. However, the majority of gearbox related fps is caused by the described signal drifts due to normal wear in gearbox bearings (compare Figure 9 $a$)). At the same time around two thirds of CPs are correctly detected in gearbox related signals, often representing major changes such as drop in gear-oil pressure after a maintenance (compare Figure 8 $a$)). For the generator related signals the main cause of fns are relatively short temporary

changes, such as the temporary high temperatures in a generator bearing displayed in 9 $b$). The same reason causes the majority of fns in pitch related signals. In fact, 35 out of the 40 false negatives were of the kind shown in Figure 11 $c$). These shifts in pitch motor winding temperature signals were caused by systematic communication problems. At the same time many shifts were distinct enough to be detected which explains the high number of tps in pitch related signals. It can be summarised that signal drifts due to normal wear, temporary changes and the coexistence of CPs with different significance levels represent

challenges which have to be addressed in the future. Nevertheless, the algorithm gives reasonable results and was able to identify the majority of CPs present in the signals. The detailed results per component of all configurations by penalty and time horizon can be found in Appendix C.

### 5.3   Discussion of pre-processing, results, and application

From the presented results it can be concluded that Laplacian kernels in combination with bandwidths chosen based on the

batch-std heuristic are best suited for the problem at hand. This configuration in combination with cost-based penalties clearly outperformed all other configurations. Analysis showed that correctly classified CPs are often characterized by a permanent





**Table 2.** Automated signal selection results per component for best configuration (Laplace / std-max / $\alpha_{cost} = 150$) on 2 year signals.

| Component | tn | tp | fn | fp | accuracy | precision | recall | f1-score |
|---|---|---|---|---|---|---|---|---|
| Gearbox | 40 | 60 | 35 | 18 | 0.65 | 0.769 | 0.632 | 0.694 |
| Generator | 162 | 22 | 24 | 7 | 0.86 | 0.759 | 0.478 | 0.587 |
| Pitch system | 98 | 82 | 39 | 8 | 0.79 | 0.911 | 0.678 | 0.777 |
| Electrical system | 24 | 3 | 6 | 1 | 0.79 | 0.75 | 0.333 | 0.462 |
| Ambient conditions | 30 | 16 | 0 | 0 | 1 | 1.0 | 1.0 | 1.0 |
| Others | 50 | 12 | 6 | 3 | 0.87 | 0.8 | 0.667 | 0.727 |
| Total | 404 | 195 | 110 | 37 | 0.8 | 0.841 | 0.639 | 0.726 |

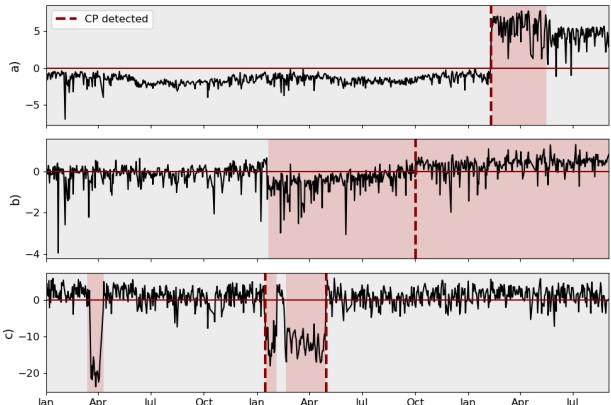

**Figure 11.** Examples of partially correct classified oil temperature (a), gearbox bearing temperature (b) and pitch motor winding temperature (c). Change in background color indicates true CPs, dashed lines detection.

nature, sharp transitions between states and a significant difference in relation to the regular signal noise. The latter two qualities are particularly amplified by the pre-processing procedure (compare section 4.1). In order to demonstrate its importance the algorithm was run on the database with only a minimum of signal pre-processing, namely a daily averaging of the measurements in order ensure computational feasibility. Results show a drop in F1-scores from 0.83 to 0.6 for validation of the 2 year signals and an even more dramatic decline from 0.73 to 0.27 for the selection task. This highlights the pre-processing procedure as an essential part of the approach. The detailed results of the run without pre-processing can be found in Appendix D1.

However, differentiated considerations are required to adequately interpret the presented results. While the algorithm is able judge the signals with an accuracy of at least 80% across all evaluation objectives there is a significant difference in F1-scores between automated signal validation and training data selection. The clear maximum of F1-scores for the validation of 2 year signals suggests that this is the application the algorithm is suited best for, but not limited to. Analysis has shown that the





**Table 3.** Conservative penalty choices and their performance for Laplace kernels and batch-std.

| evaluation objective | penalty-values | time horizon | tn | tp | fn | fp | accuracy | precision | recall | f1-score |
|---|---|---|---|---|---|---|---|---|---|---|
| validation | $\alpha_{cost} = 200$ | 2 years | 412 | 113 | 69 | 4 | 0.88 | 0.966 | 0.621 | 0.756 |
| validation | $\alpha_{cost} = 130$ | 1 year | 972 | 107 | 109 | 10 | 0.9 | 0.914 | 0.495 | 0.643 |
| selection | $\alpha_{cost} = 200$ | 2 years | 412 | 142 | 162 | 13 | 0.76 | 0.916 | 0.467 | 0.619 |
| selection | $\alpha_{cost} = 130$ | 1 year | 972 | 125 | 182 | 17 | 0.85 | 0.88 | 0.407 | 0.557 |

reduction in performance is predominantly caused by a few challenges common across signals. One of them being fps due to drifts induced by normal wear. A trend-removal step in the pre-processing procedure is suggested to mitigate the effect of regular wear. The challenge of multiple CPs with different significance levels can be tackled by an iterative application of the algorithm to the automatically selected training sub-sequenced. In fact, the results from the two different time horizons have

shown that by dividing the changes of different significant levels into two sub-signals each can be detected successfully. Lastly, the impact of temporary changes on NBM training depends on the significance of the change as well as on the duration of its presence. Short and significant temporary changes can be removed with existing statistical filtering approaches (compare e.g. Bangalore et al. (2017)). A combined application with the presented CPD algorithm is recommended. These measures will help to improve the performance of the algorithm in an application scenario beyond the presented results.

A more conservative approach would be to aim for maximal precision instead of maximal F1-scores. This corresponds to minimal training data loss while still identifying the most significant CPs. As an example, Table 3 shows the algorithm's results for conservative cost-based penalty factors 50 points above the optimal F1-Scores for the Laplace kernel configuration across the different time horizons and objectives. The remaining few fps can be exclusively attributed to normal wear phenomena like shown in Figure 9 $a$) which can be a useful indicator by itself in a NBM setting, as discussed before. This means that without

significant loss in training data the algorithm is able to identify and correctly flag the 62%/50% most severe cases among the affected signals. When automatically selecting training data with these conservative penalty-values the 44%/41% most severe CPs are automatically excluded. For illustration, the CPs depicted in Figure 3 as well as the successful detections depicted in Figures 8 and 11 were all correctly identified with the conservative penalty factors. Therefore, the method shows a clear advantage over classical pre-processing procedures.

An alternative and potentially even more effective way to apply the algorithm in the context of NBM is to run it directly on the training error once a model is considered well trained. Conceptually it is clear that CPs in the model input or target induce CPs in the model error. In fact, any CP in the model training error represents a change in conditions the model was not able to adapt to and is therefore worth investigating. The presented pre-processing procedure itself exposes similarities with early approaches of NBM when simple linear models with basic SCADA inputs were used (compare e.g. Schlechtingen and Santos

(2011)). This suggests that an application to the training error should be effective and the hyperparameter suggestions from this study applicable. However, these assumptions need to be confirmed with further experiments. A disadvantage of the training





error based approach is that it requires computational expensive model training before validation of the training period. In fact, a combination of both approaches might be the best practice.

## 6 Summary and Outlook

Literature points out systematic changes in sensor behaviour as one of the most severe challenges when analysing wind turbine

SCADA data for early failure detection. This is due to the fact that most approaches require a clean baseline data set to fit their respective models. This study therefore systematically analysed and, for the first time, quantified the presence of CPs in wind turbine SCADA data. 600 signals from 33 Turbines were analysed for an operational period of more than 2 years. During this time one third of the signals showed one or more significant changes in behaviour induced by sensor and component malfunctions or maintenance actions. This finding highlights the need of an automated CP detection method. A kernel-based offline CP

detection algorithm was introduced which consists of a normalising pre-processing procedure and recommendations on how to choose a number of crucial hyperparameters. Performance of the algorithm was evaluated across Linear, Gaussian and Laplace kernel configurations, different kernel-bandwidths, and penalty selection schemes. Laplace kernels in combination with newly introduced heuristics for bandwidth and penalty selection performed best and clearly outperformed existing alternative approaches. Signals containing a CP were labelled as such with a F1-score of up to 0.86 which translates into approximately

50 misclassifications among the 600 analysed signals. Evaluation on a per-CP basis resulted in a maximum F1-score of 0.73. Despite the reduction in performance the algorithm was able to automatically exclude the most significant 40% to 60% of all true CPs without significant loss of training data. Therefore, the presented algorithm represents a valuable tool for SCADA data pre-processing and will help data driven methods to become more robust despite widely spread data quality issues. Future research has to confirm the presented results for different SCADA data sets. Moreover, an extension of signal pre-processing,

an iterative application of the algorithm and the combination with existing statistical filtering methods hold the potential for further improve performance. Further development is encouraged by making the code available under the GNU general public license.

*Code and data availability.* The code of the kernel based CPD-algorithm publicly available (see Letzgus (2020)). The SCADA data-set used during this study is propreatary but a number exemplary pre-processed signal samples are published along with the code.





## Appendix A: CP summary statistics for 1-year signals

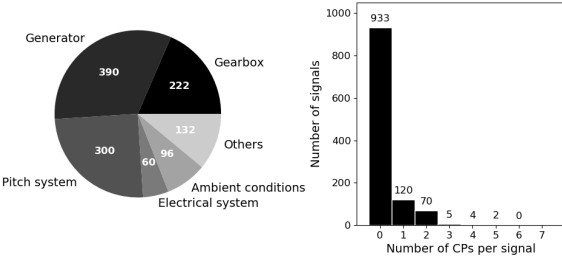

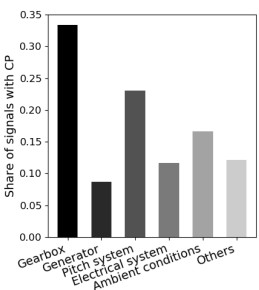

**Figure A1.** Number of signals per component (left), number of CPs per signal (center), and share of signals with CPs per component (right) for the 1-year time horizon.





## Appendix B: List of analysed signals

**Table B1.** Full list of analysed signals

| Component | Signal | Number of signals |
|---|---|---|
| Gearbox | Gear Bearing Temperature | 48 |
| Gearbox | Gearbox Temperature | 18 |
| Gearbox | Gear Oil Temperature | 30 |
| Gearbox | Gear Oil Pressure | 15 |
| Generator | Generator Bearing Temperature | 66 |
| Generator | Generator Winding Temperature | 81 |
| Generator | Cooling Temperature | 48 |
| Pitch system | Pitch Converter Temperature | 90 |
| Pitch system | Pitch Motor Temperature | 45 |
| Pitch system | Hydraulic Oil Temperature | 15 |
| Electrical system | Transformer Temperatures | 15 |
| Electrical system | Box Temperatures | 15 |
| Ambient conditions | Ambient temperature | 33 |
| Ambient conditions | Tower Temperature | 15 |
| Others | Shaft Bearing Temperature | 18 |
| Others | Nacelle Temperature | 33 |
| Others | Rotor Break Temperature | 18 |
| Total | | 600 |





## Appendix C: Detailed results per component

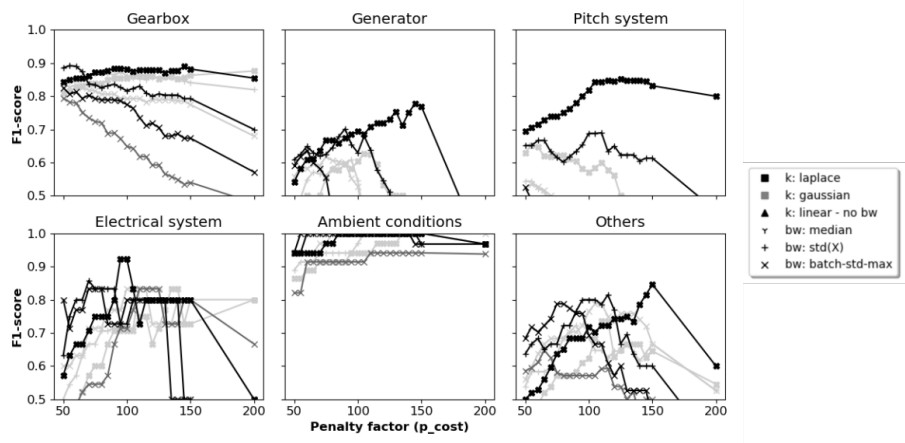

**Figure C1.** 2-year signal validation: F1-Scores per component for different hyperparameter configurations and penalty values

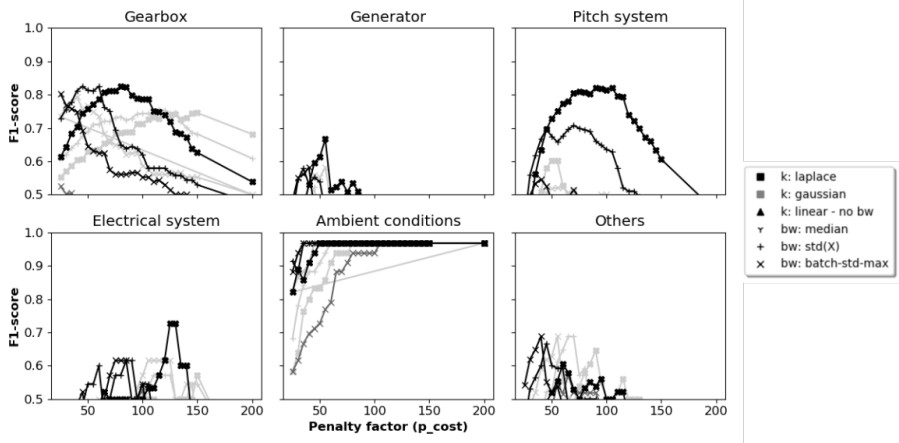

**Figure C2.** 1-year signal validation: F1-Scores per component for different hyperparameter configurations and penalty values.





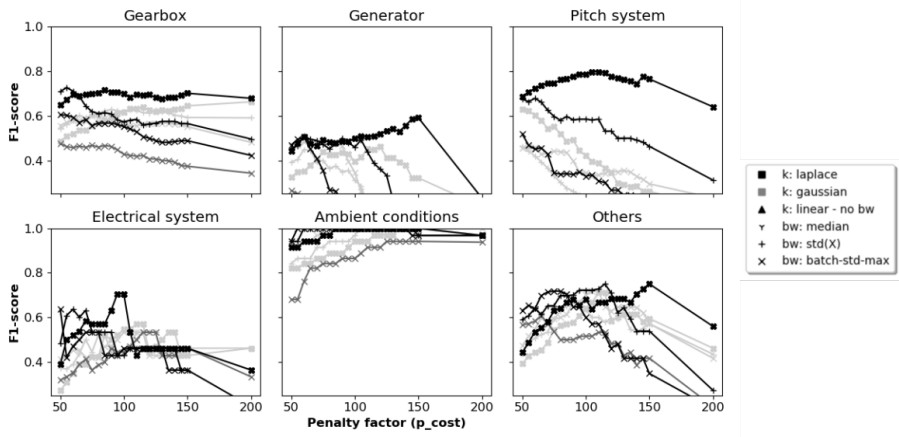

**Figure C3.** 2-year signal selection: F1-Scores per component for different hyperparameter configurations and penalty values.

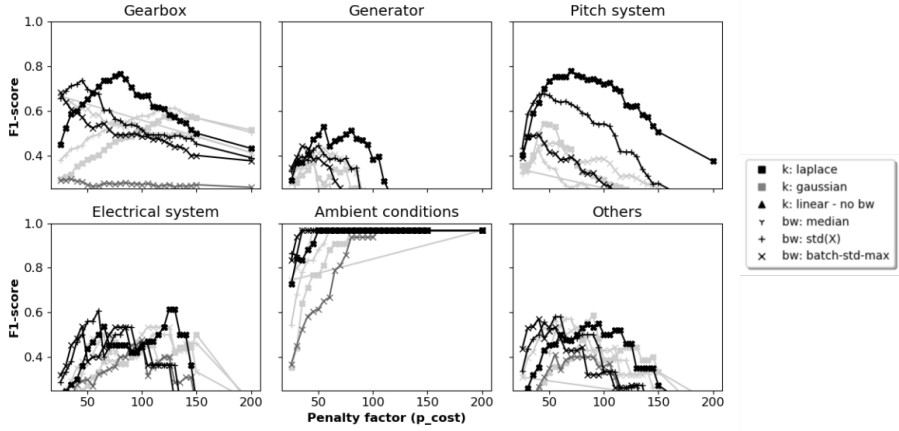

**Figure C4.** 1-year signal selection: F1-Scores per component for different hyperparameter configurations and penalty values.



**Table C1.** Results validation per component for best configuration (Laplace / std-max / $\alpha_{cost} = 80$) on 1 year signals.

| Component | tn | tp | fn | fp | accuracy | precision | recall | f1-score |
|---|---|---|---|---|---|---|---|---|
| Gearbox | 128 | 66 | 8 | 20 | 0.874 | 0.767 | 0.892 | 0.825 |
| Generator | 349 | 15 | 19 | 7 | 0.933 | 0.682 | 0.441 | 0.536 |
| Pitch system | 220 | 54 | 15 | 11 | 0.913 | 0.831 | 0.783 | 0.806 |
| Electrical system | 45 | 5 | 2 | 8 | 0.833 | 0.385 | 0.714 | 0.5 |
| Ambient conditions | 78 | 15 | 1 | 0 | 0.989 | 1.0 | 0.938 | 0.986 |
| Others | 110 | 8 | 8 | 6 | 0.894 | 0.571 | 0.5 | 0.533 |
| Total | 930 | 163 | 53 | 52 | 0.91 | 0.758 | 0.755 | 0.756 |

**Table C2.** Results selection per component for best configuration (Laplace / std-max / $\alpha_{cost} = 80$) on 1 year signals.

| Component | tn | tp | fn | fp | accuracy | precision | recall | f1-score |
|---|---|---|---|---|---|---|---|---|
| Gearbox | 128 | 77 | 18 | 29 | 0.81 | 0.726 | 0.811 | 0.766 |
| Generator | 349 | 21 | 27 | 13 | 0.9 | 0.618 | 0.438 | 0.512 |
| Pitch system | 220 | 87 | 33 | 24 | 0.84 | 0.784 | 0.725 | 0.753 |
| Electrical system | 45 | 5 | 4 | 8 | 0.81 | 0.385 | 0.556 | 0.455 |
| Ambient conditions | 78 | 15 | 1 | 0 | 0.99 | 1.0 | 0.938 | 0.968 |
| Others | 110 | 9 | 10 | 6 | 0.88 | 0.6 | 0.474 | 0.529 |
| Total | 930 | 214 | 93 | 80 | 0.87 | 0.728 | 0.697 | 0.712 |



## Appendix D:  Results of algorithm without pre-processing

**Table D1.** Performance of the algorithm without pre-processing on 2 year signals.

| evaluation objective | penalty | time horizon | tn | tp | fn | fp | accuracy | precision | recall | f1-score |
|:---:|:---:|:---:|:---:|:---:|:---:|:---:|:---:|:---:|:---:|:---:|
| validation | $\alpha_{cost} = 7$ | 2 years | 264 | 155 | 38 | 172 | 0.67 | 0.47 | 0.8 | 0.6 |
| selection | $\alpha_{cost} = 30$ | 2 years | 376 | 101 | 212 | 357 | 0.46 | 0.22 | 0.32 | 0.26 |





*Competing interests.*  The author declares no competing interests.

*Acknowledgements.*  The author greatly acknowledges support by the Berlin International Graduate School in Model and Simulation based Research (BIMoS) and the Open Access Publication Fund of TU Berlin. Moreover, the author thanks Greenbyte AB, in particular Pramod Bangalore, for the cooperation and the fruitful discussions.



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
