# Peer review of "Change-point detection in wind turbine SCADA data for robust condition monitoring with normal behaviour models"

_Wind Energy Science, 2020_

## Referee Comment (RC1) · Anonymous Referee #1 · 21 May 2020

The paper represents a very valuable contribution to the processing of SCADA data for condition monitoring by automatically identifying change points in the time series. The paper is generally suitable for publication, though there are a few points to be addressed: Page 7, line 29: what is the bracketed (1) meant to represent? Page 8, line 4: what is P? In general, ensure that al symbols in equations are properly explained. Page 12, line 17: should the maximum std not be evaluated over k values rather than i? In equation 10, what are D and T? Page 14, line 14: should the statement 'whereas splitting the might require detection of a less severe change in one half of the signals' be something like 'whereas splitting them might result in only the detection of a less severe change in one half of the signal time series'. Section 5.2: I am not sure I understand the

analysis by CP – who exactly are the time series split? I assume there is one section of data with a CP within it, but where is the split made? Page 17, line 1: it is said that there are two CPs in 11a, but the figure shows only one shaded region Page 17, line 14: the reference to 'signal drifts' should be changed to 'signal changes'. A drift suggests a problem with the sensor itself, whereas what is suggested is that the temperature change is genuine but just due to bearing wear. Page 19, line 2: why would you want to remove such a trend? I assume this technique could be used to identify 'clean' sections of data which can then be analysed with fault detection algorithms. Taking out the trend would then be counter-productive A general point: although CPs are most likely due to some change in set points, maybe after maintenance for example, many could be considered as genuine faults (a sensor drift or comms problem is a fault after all). Perhaps the authors could comment on how to differentiate between CPs which are related to faults and those which are not as they need to be treated different in such as NBM. Some typos: Page 3, line 17: 'multiple' Page 3, line 21: space between 'specifies' and 'the' Page 4, line 19: better to be explicit in terms of 'oil pressure' (not just pressure which could be atmospheric pressure) Page 7, line 25: 'let us' rather than 'let's' (avoid contractions in formal writing) Page 8, line 24: 'calculation' Page 9, line 23: 'automated' Page 10, line 8: 'occurrence' Page 13, line 13: should be 'based on' Page 13, line 22: 'reversely' should be 'conversely' Page 13, line 28: 'algorithm's' Page 18, line 4: should be 'to ensure' Page 18, line 8: should be 'able to'

---

## Referee Comment (RC2) · Anonymous Referee #2 · 22 Jul 2020

The paper represents an valuable addition to the automatic data processing of SCADA data By detecting the CPs reliably for further application of SCADA, such as monitoring and fault detections.

General question to the manuscript.

-In the manuscript it is mentioned that data 33 wind turbines from 3 different sites are analyzed in this paper. Do they represent the same turbine typology, i.e. geared versus direct drive, synchronous generator versus DFIG etc. What kind of site conditions they represent, complex terrain versus flat terrains. Age of the wind turbines? It is important in my opinion to discuss the representativeness of the data that are used in the analysis

as the Resulting method will be applied to different turbine types, ages, site conditions etc.

-the data analyzed here are temperature data according to the table B.1 What is the reason behind this choice. Are there vibration data from the wind turbine also available for the analysis. Will the algorithm change if other Type of sensors are analyzed, e.g. acceleration data.

-Change of operation modes. Do the algorithm consider changes in the operational state of the wind turbines? For example, down regulation of power due to grid demand, noise reduced operation due to noise regulation In the night with medium high wind speeds. These can looks like CPs in the data possibly.

-minor comments and edits can be found in the attached PDF file

Please also note the supplement to this comment:
https://wes.copernicus.org/preprints/wes-2020-38/wes-2020-38-RC2-supplement.pdf

———————————————————

[Figure]

**Supplement:**

[revised manuscript text omitted]

---

## Author Comment (AC1) · 10 Aug 2020

**Referee#1: Please find below the answers to the individual remarks ordered from general to specific.**

**Q1:** A general point: although CPs are most likely due to some change in set points, maybe after maintenance, for example, many could be considered as genuine faults (a sensor drift or comms problem is a fault after all). Perhaps the authors could comment on how to differentiate between CPs which are related to faults and those which are not as they need to be treated differently in such as NBM.

**A1:** The distinction between CPs induced by faults and CPs caused by other effects is indeed difficult and without additional information, such as SCADA log-files or maintenance reports, speculative to some degree. In such a case, the only indication we see lays in the CP characteristics itself (as discussed in chapter 2.2). Firstly, changes in signal behaviour can be classified as being permanent or temporary. The latter ones consist of two CPs, where signal behaviour returns to its original pattern after a limited period (not longer than an internal of periodic inspections). For such changes, commonly, the first CP was caused by a malfunction, therefore being fault-related, which was consecutively corrected. The corrective action induced the second, non-fault related CP. A permanent change in signal behaviour, on the other hand, is not reverted. Permanent changes are more likely to be attributed to be maintenance- and therefore non-fault-related. However, there still is the possibility of the change being induced by a fault which has not been discovered or was judged not to be severe enough to be fixed. Another distinction can be made between gradual and abrupt changes. Gradual changes can almost exclusively be attributed to be fault-related whereas abrupt changes could be either. Lastly, some physics of failure considerations might help to correctly identify the nature of an observed change. For temperature measurements, for instance, changes that manifest themselves in overall higher temperatures are more likely to be attributed to failures whereas changes leading to lower temperatures are more likely to be attributed to maintenance actions. For sensors like oil-pressure measurements, the exact opposite would be the case. Taking all these criteria together should enable the analyst to make an informed guess about the nature of the observed change, although some uncertainty remains. We think that the reviewer's question raises a point worth discussing and therefore incorporated this line of thought into the restructured section 2 (revised manuscript p. 4, line 9 ff. and p. 5, line 3 ff.).

With respect to the application of NBMs we would argue that the impact of fault and non-fault related CPs depends on the concrete question to be addressed. For the provision of clean training data, the main practical issue faced in real-world NBM application and therefore the focus of our contribution, both kinds of CPs have the same distorting effect on model training. Thus, any kind of CP violates the central assumption of NBMs and consequently has to be removed from the training data sets to ensure the method's feasibility.

**Q2:** Section 5.2: I am not sure I understand the analysis by CP – who exactly are the time series split? I assume there is one section of data with a CP within it, but where is the split made?

**A2:** Section 5 presents the performance evaluation of the algorithm. Performance is evaluated for the 600 selected signals, each covering two years of operation. Additionally, each of the signals is split exactly in the middle, resulting in 1200 signals, each covering one year of operation. With this split, a two-year signal, that contains only one CP indeed results in one signal with and one without a CP, each of length one year. In case multiple CPs are present, they might end up in either of the two shorter

signals, depending when they occurred (first half/second half). The evaluation for these 1200 shorter signals was conducted for three main reasons:

i. To demonstrate the method's applicability to SCADA signals of different length. We think that this generalization property is an essential feature.

ii. Previous work of the authors has shown that at least one year of SCADA data is required to train robust NBMs and which is in line with other publications explicitly recommending training data covering all four seasons (compare [1] and [2]). Having in mind the application of providing clean training data sets we think that demonstration on the one-year signals is closer to the application setting and therefore valuable.

iii. The experiment showed that in many cases less dominant CPs could be successfully detected when a dominant CP was removed by the splitting procedure, which inspired the idea of an iterative CP removal (as discussed in section 5.3).

Thanks to the referee's remark we realised that the splitting procedure might not have been motivated adequately. This has been updated for the revised manuscript in section 2. There the split is initially discussed regarding its impact on the CP statistics. All three points mentioned above were explicitly incorporated (compare revised manuscript p. 6 lines 16 ff.).

**Q3:** Page 19, line 2: why would you want to remove such a trend? I assume this technique could be used to identify 'clean' sections of data which can then be analysed with fault detection algorithms. Taking out the trend would then be counter-productive

**A3:** In the given context the removal of overall signal trends is suggested only for signal pre-processing as part of the CP detection process. The data used for NBM training and application would still contain the trend but the training periods would be selected based on the outcome of the CP algorithm. The reasoning behind the trend-removal suggestion is that a steady trend which is present throughout the observed period does represent a shift in the signal's distribution but this steady shift itself is not changing and therefore should not be flagged as a CP.

This being said, the distinction between rising temperatures due to normal wear, which an NBM then would have to account for as 'normal' and an increased wear leading up to an early end of component life might be difficult. To our knowledge has not been addressed in literature so far and would be an interesting point for further research, since the presence of trends in the training data has been reported to be potentially indicative for slowly developing component problems (compare [3]).

**Q4:** Page 17, line 1: it is said that there are two CPs in 11a, but the figure shows only one shaded region.

**A4:** Shaded regions represent homogeneous periods with no change-points. True change-points are then indicated by the change in background colour. In figure 11a) the two true change points are in February (background colour changes from grey to red) and May (background colour changes from red to grey) of the second depicted year of operation. This way of visualizing the results was chosen to ensure both types of CPs, true and detected, are visible also in case of an exact detection where they overlap (compare the first CP in figure 11a). The figure captions have been updated to enhance clarity. *'Change in background colour indicates true CPs, dashed lines detection.'* was replaced by *'Each change in background colour indicates a true CP, each dashed line indicates a detected CP'.*

**Q5:** Page 14, line 14: should the statement 'whereas splitting them might require detection of a less severe change in one half of the signals' be something like 'whereas splitting them might result in only the detection of a less severe change in one half of the signal time series.

**A5:** The sentence in question was changed to ensure comprehensibility: *'Secondly, in case the two-year signal contains multiple CPs, detection of only the most significant one is enough for the signal to be evaluated as correctly classified (TP). When splitting this two-year signal into two one-year signals to analyse and evaluate them separately, detection of a less severe change in one of the signals might be required for both signals to be evaluated as correctly classified (both TP).'* (compare revised manuscript page 14, lines 19 ff.).

**Q6:** In general, ensure that all symbols in equations are properly explained.

   i.   **Q6.1:** Page 7, line 29: what is the bracketed (1) meant to represent?

   **A6.1:** This was meant to be a reference to Equation (1) which was therefore corrected to '(*compare Eq. (1))*'.

   ii.  **Q6.2:** Page 8, line4: what is P?

   **A6.2:** P stands for the penalty term which acts as a regulariser for model complexity. For clarification, the missing reference was added inline as follows (bold): '*Therefore, a regularisation term P(τ) was proposed for example by Lavielle (2005) which penalises for every additional CP and therefore reduces the complexity of the segmentation (compare Eq. (2))*'.

   iii. **Q6.3:** Page 12, line 17: should the maximum std not be evaluated over k values rather than i?

   **A6.3:** We agree with the reviewer and have updated the manuscript accordingly.

   iv.  **Q6.4:** In equation 10, what are D and T?

   **A6.4:** T stands for the total number of time-steps the signal consists of, as defined in section 3.1. However, we agree that this should be stated again in proximity to Equation 10. The naming of D, which here stands for the actual number of segments, was named N in the earlier problem formulation in section 3.1 (compare Equation (1)). This inconsistency was corrected accordingly. Moreover, we noticed that the choice of D within this publication was not reported. The paragraph before equation 10 was therefore updated accordingly (compare revised manuscript page 13, lines 8 ff.).

**Q7:** Page 17, line 14: the reference to 'signal drifts' should be changed to 'signal changes'. A drift suggests a problem with the sensor itself, whereas what is suggested is that the temperature change is genuine but just due to bearing wear.

 **A7:** We agree with the reviewer and have updated the manuscript accordingly.

**Q8:** Some typos:

    i. Page 3, line 17: 'multiple'
    ii. Page 3, line 21: space between 'specifies' and 'the'
    iii. Page 4, line 19: better to be explicit in terms of 'oil pressure' (not just pressure which could be atmospheric pressure)
    iv. Page 7, line 25: 'let us' rather than 'let's' (avoid contractions in formal writing)
    v. Page 8, line 24: 'calculation'
    vi. Page 9, line 23: 'automated'
    vii. Page 10, line 8: 'occurrence'
    viii. Page 13, line 13: should be 'based on'
    ix. Page 13, line 22: 'reversely' should be 'conversely'
    x. Page 13, line 28: 'algorithm's'
    xi. Page 18, line 4: should be 'to ensure'
    xii. Page 18, line 8: should be 'able to'

**A8:** We agree with the reviewer and have updated the manuscript accordingly.

**REFERENCES:**

[1]     Bach-Andersen, M., Rømer-Odgaard, B., and Winther, O.: Flexible non-linear predictive models for large-scale wind turbine diagnostics, Wind Energy, 20, 753–764, 2017.

[2]     Letzgus S., Training data requirements for SCADA based condition monitoring using artificial neural networks, 2019, EAWE PhD Seminar, Nantes

[3]     Letzgus S, SCADA-based anomaly detection – challenges for automated application of artificial neural networks, 2018, EAWE PhD Seminar,  Bruxelles

---

## Author Comment (AC2) · 10 Aug 2020

**Referee#2: Please find below the answers to the individual remarks ordered from general to specific.**

**Q1:** In the manuscript, it is mentioned that data 33 wind turbines from 3 different sites are analysed in this paper. Do they represent the same turbine typology, i.e. geared versus direct drive, synchronous generator versus DFIG etc.? What kind of site conditions they represent, complex terrain versus flat terrains? Age of the wind turbines? It is important in my opinion to discuss the representativeness of the data that are used in the analysis as the Resulting method will be applied to different turbine types, ages, site conditions etc... .

**A1:** The turbines are from different manufacturers and all of them are geared and equipped with DFIGs. All turbines were commissioned later than 2013 and the analysed periods fall within the first five years of operation. The sites can be characterised as moderately complex with mild elevation changes and occasional vegetation. This information was indeed missing and will be incorporated in section 2.1 of the revised manuscript accordingly (revised manuscript: p. 4, line 14 ff.).

Even though the turbines represent a rather homogeneous set we expect the method to perform equally well on temperature measurements along the drive train from turbines with different configurations where different sensors might be in place. This is due to the method's good performance over the wide range of different temperature signals as well as the different characteristics of the detected change-points. Neither do we expect the method's performance to decrease for older turbines or turbines in different site conditions. However, these characteristics might influence the presented cp statistics, with older turbines or turbines exposed to higher loads showing an increased amount of change-points due to increased wear and consecutive maintenance actions. This line of thought was added to section 2 as well: 'Even though these findings might vary across different turbine types, ages and site conditions the order of magnitude of CP presence highlights the necessity of a robust CPD methodology.' (compare revised manuscript p. 6, line 14 ff.).

**Q2:** The data analysed here are temperature data according to the table B.1 What is the reason behind this choice?

**A2:** In SCADA based monitoring of wind turbines using NBMs two approaches can be distinguished - performance and temperature monitoring. The former aims to detect abnormal deviations from the turbines usual power output, whereas the latter aims to detect deviations from the healthy thermal equilibrium conditions. Although both approaches have proven to be valuable (particularly in combination) temperature monitoring is better suited for detecting malfunctions in the components along the drive train, which account for the majority of turbine downtime (compare [1]). Moreover, the challenge of change-points in wind turbine SCADA data was mainly reported in the context of temperature monitoring in literature. Therefore, we decided to focus on temperature data. Nevertheless, the methods performance over a wide range of different temperature signals as well as over the different characteristics of the detected change-points suggests that the method can potentially be extended to other signals found in SCADA systems, a proposition that has been incorporated into the outlook section 6 of the revised manuscript (compare revised manuscript p.21, lines 11 ff.). Thanks to the referee's comment it also became clear, that neither the distinction between temperature and performance monitoring nor our motivation for variable selection were stated explicitly enough. Therefore, they incorporated into the introductory section (compare

updated manuscript p. 2, line 21) as well as the data set description of section 2 (compare revised manuscript p. 4, 18ff.).

**Q3:** Are there vibration data from the wind turbine also available for the analysis. Will the algorithm change if other types of sensors are analysed, e.g. acceleration data?

**A3:** Vibration/Acceleration data were not available for this study. Nevertheless, we assume that in principle the suggested kernel-based change-point detection algorithm should also be useful to analyse measurements from these kinds of sensors. [2] for example presents experimental results of kernel-based change-point detection being successfully applied to the segmentation of audio signals. In terms of structure and time resolution, audio signals are much closer to vibration data than the SCADA data analysed in this study. One particular challenge we see at this point is that the high data resolution could impose numerical challenges for computing the respective gram-matrix. In any case, the proposed data pre-processing method would need to be adjusted to the different types of data and could potentially help to overcome these problems. We think this is an interesting question that could be addressed in the future and therefore incorporated it into the outlook section 6 (compare revised manuscript p. 21, 10 ff.).

**Q4:** Change of operation modes. Does the algorithm consider changes in the operational state of the wind turbines? For example, downregulation of power due to grid demand, noise-reduced operation due to noise regulation in the night with medium/high windspeeds. These can look like CPs in the data possibly.

**A4:** The proposed framework considers changes in operational states of the wind turbine in two different ways. Firstly, the pre-processing procedure acts as a normalization which puts the measured temperature in relation to the operational state. Secondly, by averaging the signals over a full day, which was originally motivated by computational considerations, the impact of such presumable sub-day events is further reduced. This helps the algorithm to focus on the most significant and long-lasting changes and is part of the reason, why the pre-processing has such a crucial effect on the algorithm's performance (compare section 5.3).

**Q5:** Minor comments and edits can be found in the attached PDF file:

  i.   **Q5.1:** The pre-processing takes care of seasonal effects. What about diurnal effects?

       **A5.1:** To reduce the numerical effort of computing the gram-matrix a daily averaging of the signals is part of the pre-processing procedure. This also removes all diurnal effects. Moreover, diurnal effects would be detected only with a penalty much lower than the proposed one, since the reduction in the cost function would need to compensate for as many change-points as days in the analysed period. Seasonality on the other hand induces 2 to 4 false CPs in each seasonal signal when not handled prior to the CP optimisation step and is therefore much more likely to be flagged by the algorithm.

 ii.   **Q5.2:** Page 4, line 4: 1Hz sampling is usually possible, the only problem is they are not being stored due to data storage reasons. The second reason you don't see them is because OEMs don't give access to wind farm operators.

**A5.2:** We agree with the reviewer and have updated the manuscript accordingly (compare revised manuscript p. 4, lines 3 ff.).

   iii.     **Q5.3**: Notes on spelling/grammar in the PDF-file.

**A5.3:** We agree with the reviewer and have updated the manuscript accordingly.

**REFERENCES:**

[1] Dao, C., Kazemtabrizi, B., Crabtree, C.: Wind turbine reliability data review and impacts on levelised cost of energy, Wind Energy, 22, 1848-1871, 2019

[2] Arlot, S., Celisse, A., and Harchaoui, Z.: Kernel change-point detection, arXiv preprint arXiv:1202.3878 ,2012.